# DIFFERENTIABLE EXPECTED BLEU FOR TEXT GENERATION

## ABSTRACT

Neural text generation models such as recurrent networks are typically trained by maximizing data log-likelihood based on cross entropy. Such training objective shows a discrepancy from test criteria like the BLEU metric. Recent work optimizes expected BLEU under the model distribution using policy gradient, while such algorithm can suffer from high variance and become impractical. In this paper, we propose a new Differentiable Expected BLEU (DEBLEU) objective that permits direct optimization of neural generation models with gradient descent. We leverage the decomposability and sparsity of BLEU, and reformulate it with moderate approximations, making the evaluation of the objective and its gradient efficient, comparable to common cross-entropy loss. We further devise a simple training procedure with ground-truth masking and annealing for stable optimization. Experiments on neural machine translation and image captioning show our method significantly improves over both cross-entropy and policy gradient training.

## 1 INTRODUCTION

Text generation includes a broad set of natural language processing (NLP) tasks, such as machine translation (Sutskever et al., 2014; Bahdanau et al., 2014), dialog (Serban et al., 2016; Bordes et al., 2016), image captioning (Karpathy & Fei-Fei, 2015; Vinyals et al., 2015), and others. Recent years have seen great advances in the field, especially with the use of modern neural network models, e.g., sequence-to-sequence for neural machine translation (Sutskever et al., 2014), and efficient training with gradient back-propagation. A text generation system is usually evaluated with certain measures. BLEU score (Papineni et al., 2002) is one of the most widely-used metrics. On the other hand, however, the dominant training objective for the models is to maximize the data log-likelihood (e.g., based on the cross-entropy loss), resulting in a discrepancy between the training and test criteria.

The BLEU metric is non-differentiable, and hence disables direct gradient descent optimization. Previous efforts have been made attempting to address the issue. For example, Ranzato et al. (2015) use the policy gradient (Sutton et al., 2000) with BLEU as the reward. The algorithm maximizes the expected BLEU score under the model distribution. However, such reinforcement learning approach is known to be difficult for training due to exceedingly high variance and poor exploration efficiency. Recent work (Casas et al., 2018; Zhukov & Kretov, 2017) made preliminary attempts to develop differentiable approximations of BLEU for neural model training, but only studied on toy tasks or obtained negative results. Earlier literature (Rosti et al., 2011; He & Deng, 2012; Pauls et al., 2009; Smith & Eisner, 2006) has developed differentiable variants of BLEU designed for statistical machine translation, which are not directly applicable to neural text generation.

In this paper, we develop a new differentiable BLEU objective that is end-to-end trainable with gradient descent for neural generation models. Specifically, starting from the conventional expected BLEU objective, we reformulate it with moderate approximations, and leverage the sparsity of BLEU scores to enable efficient evaluation of the resulting new objective and its gradient w.r.t. model parameters. The computational complexity is comparable to the common cross-entropy loss. No sampling from the huge sequence space nor cumbersome policy gradient is needed. For stable and efficient optimization, we further devise a training procedure for the objective with simple ground-truth masking and annealing.

We evaluate the proposed method in the tasks of neural machine translation and image captioning, and obtained significantly improved performance and more stable convergence compared to the commonly used cross-entropy training and policy gradient.

## 2 RELATED WORK

Text generation using deep neural models such as recurrent neural networks (Sutskever et al., 2014; Mikolov et al., 2010) has achieved great progress in many concrete tasks like machine translation (Bahdanau et al., 2014; Vaswani et al., 2017). However, these models are typically trained with the maximum-likelihood objective for convenience, which can lead to sub-optimal performance due to the discrepancy between the training objective and the test metrics such as BLEU. Many works resort to reinforcement learning for direct optimization of the non-differentiable evaluation metrics. For example, Ranzato et al. (2015); Rennie et al. (2017); Liu et al. (2017); Shen et al. (2015); Smith & Eisner (2006) propose to use policy gradient or minimum risk training to optimize the expected BLEU score. A variety of training tricks are used to reduce variance and stabilize the learning.

Another line of research aims to close the discrepancy by making BLEU score differentiable. Our work falls into this category. In the modern neural text generation context, Zhukov & Kretov (2017); Casas et al. (2018) made the initial attempts to develop differentiable BLEU objectives. The key idea is to make soft approximations to the count of $n$-gram matching in the original BLEU formulation. However, their derivations are preliminary, and only toy or negative results are obtained. Our new formulation uses a couple of similar approximations or assumptions. We provide clear intuitions of leveraging the sparsity of BLEU score, and decompose the goal into multiple derivation steps. Along with the devised training procedure, to the best of our knowledge, we are the first to develop end-to-end gradient descent BLEU training for neural models, which is highly practical and achieves greatly improved results. Earlier work has proposed differentiable BLEU objectives in the context of *statistical* machine translation (Rosti et al., 2011; He & Deng, 2012; Pauls et al., 2009). For example, Rosti et al. (2011) adapts expected BLEU on confusion networks to train the weights of different features. Their context differs from the neural generation setting (e.g., they do not make differentiable approximations to the $n$-gram count) and is not directly applicable for neural model training.

## 3 DIFFERENTIABLE EXPECTED BLEU

### 3.1 BACKGROUND

We first establish notations for the sequence generation setting. Let $\boldsymbol{y} = (y_1, \ldots, y_T)$ be a candidate sequence generated by a model $p_\theta(\boldsymbol{y})$ with parameter $\boldsymbol{\theta}$. Let $\boldsymbol{y}^* = (y_1^*, \ldots, y_{T^*}^*)$ be a reference sequence (i.e., ground truth). Here $T$ and $T^*$ are the lengths of $\boldsymbol{y}$ and $\boldsymbol{y}^*$, respectively. Further define $\boldsymbol{y}_{a:b} = (y_a, \ldots, y_{b-1})$ as a sub-sequence of $\boldsymbol{y}$ that starts from index $a$ and ends at index $b-1$, which is of length $b - a$. Let $\boldsymbol{y}_{\neg a:b}$ be the remaining tokens in $\boldsymbol{y}$ excluding $\boldsymbol{y}_{a:b}$. The goal is to optimize the model parameter $\boldsymbol{\theta}$ so that the resulting samples have the maximum BLEU score against reference sequences.

**The BLEU Metric**

Let us first take a review of the BLEU metric proposed in (Papineni et al., 2002) which evaluates the overlap of $\boldsymbol{y}$ against $\boldsymbol{y}^*$. Specifically, BLEU is defined as a weighted geometric mean of $n$-gram precisions:

$$\text{BLEU} = \text{BP} \cdot \exp\left(\sum_{n=1}^N w_n \log \text{prec}_n\right) \tag{1}$$

where BP is a brevity penalty depending on the lengths of $\boldsymbol{y}$ and $\boldsymbol{y}^*$; $N$ is the maximum $n$-gram order (typically $N = 4$); $\{w_n\}$ are the weights which usually take $1/N$; and $\text{prec}_n$ is the $n$-gram precision defined as:

$$\text{prec}_n = \frac{\sum_{\boldsymbol{s} \in \text{gram}_n(\boldsymbol{y})} \min\left(C(\boldsymbol{s}, \boldsymbol{y}), C(\boldsymbol{s}, \boldsymbol{y}^*)\right)}{\sum_{\boldsymbol{s} \in \text{gram}_n(\boldsymbol{y})} C(\boldsymbol{s}, \boldsymbol{y})} \tag{2}$$

where $\text{gram}_n(\boldsymbol{y})$ is the set of unique $n$-gram sub-sequences of $\boldsymbol{y}$; and $C(\boldsymbol{s}, \boldsymbol{y})$ is the number of times a gram $\boldsymbol{s}$ occurs in $\boldsymbol{y}$.

The conventional formulation above enumerates over unique $n$-grams in $\boldsymbol{y}$. However, for the derivations in the sequel, it is more convenient to enumerate over token indexes. To this end, for each $n$-gram of $\boldsymbol{y}$ starting from index $i$, namely, $\boldsymbol{y}_{i:i+n}$, we re-write the count $C(\boldsymbol{y}_{i:i+n}, \cdot)$ as follows:

$$
\begin{aligned}
C(\boldsymbol{y}_{i:i+n}, \boldsymbol{y}) &= \sum\nolimits_{i'=1}^{T-n+1} \mathbb{1}[\boldsymbol{y}_{i':i'+n} = \boldsymbol{y}_{i:i+n}] \triangleq v_{n,i}, \\
C(\boldsymbol{y}_{i:i+n}, \boldsymbol{y}^*) &= \sum\nolimits_{j'=1}^{T^*-n+1} \mathbb{1}[\boldsymbol{y}^*_{j':j'+n} = \boldsymbol{y}_{i:i+n}] \triangleq v^*_{n,i}.
\end{aligned}
\tag{3}
$$

Eq.(2) is then re-written with $v_{n,i}$ and $v^*_{n,i}$ as:

$$
\text{prec}_n = \frac{1}{T-n+1} \sum\nolimits_{i=1}^{T-n+1} \min\left(1, \frac{v^*_{n,i}}{v_{n,i}}\right) \triangleq \frac{1}{T-n+1} \sum\nolimits_{i=1}^{T-n+1} o_{n,i}
\tag{4}
$$

**Conventional Learning Methods**

As the BLEU metric is not differentiable for direct optimization, to train the model $p$, the simplest algorithm is instead to maximize the data log-likelihood $\log p(\boldsymbol{y}^*)$ which has a discrepancy from the BLEU metric we aim to maximize. To address the discrepancy, a common approach is the policy gradient algorithm (Sutton et al., 2000; Ranzato et al., 2015) that maximizes the expected BLEU:

$$
\mathcal{L}_{PG}(\boldsymbol{\theta}) = \mathbb{E}_{p_\theta(\boldsymbol{y})}\left[\text{BLEU}(\boldsymbol{y}, \boldsymbol{y}^*)\right].
\tag{5}
$$

The above expectation is intractable due to the large space of sequences. Thus the optimization has to resort to stochastic approximation, leading to gradient:

$$
\nabla_\theta \mathcal{L}_{PG}(\boldsymbol{\theta}) = \mathbb{E}_{\boldsymbol{y} \sim p_\theta(\boldsymbol{y})}\left[\text{BLEU}(\boldsymbol{y}, \boldsymbol{y}^*) \cdot \nabla_\theta \log p_\theta(\boldsymbol{y})\right].
\tag{6}
$$

However, the update is still impractical due to its exceedingly high variance, and in practice many stabilization techniques would be required (Ranzato et al., 2015).

## 3.2 THE DEBLEU OBJECTIVE

Note that in the above expected BLEU objective (Eq.5) a sequence $\boldsymbol{y}$ with $\text{BLEU}(\boldsymbol{y}, \boldsymbol{y}^*) = 0$ does not contribute to the model learning. Inspired from this, we reformulate the BLEU metric with moderate approximations, and leverage the *decomposability* and *sparsity* of BLEU. That is: a) Based on the definition (Eq.1), the BLEU evaluation effectively decomposes the whole sequence space into $n$-gram spaces, with $n$ up to $N$ (=4, typically); b) Only a small set of $n$-gram values are effective, i.e., with non-zero contributions to the final BLEU score. The resulting approximated reformulation of the expected BLEU, as well as its gradient w.r.t. $\boldsymbol{\theta}$, is directly tractable. We thus call the new objective the *differentiable expected BLEU* (DEBLEU).

We now derive DEBLEU in detail. In the sequel we omit the subscript $\boldsymbol{\theta}$ of $p_\theta$ for notation simplicity. Starting from the original expected BLEU objective (Eq.5) and the BLEU definition (Eq.1), we first make a couple of approximations for tractability. Specifically, during decoding at training time, we set the length of $\boldsymbol{y}$ to be the same of the ground truth $\boldsymbol{y}^*$, namely $T = T^*$. This assumption has also been used in previous work (e.g., Yang et al. (2018)). The brevity penalty term BP is then independent of $\boldsymbol{y}$. Secondly, as in (Zhukov & Kretov, 2017), we approximate the expectation by swapping it with other operations:

$$
\mathbb{E}_{p(\boldsymbol{y})}\text{BLEU} = \mathbb{E}_{p(\boldsymbol{y})}\text{BP} \cdot \prod\nolimits_{n=1}^{N} \text{prec}_n^{w_n} \approx \text{BP} \cdot \prod\nolimits_{n=1}^{N} \left(\mathbb{E}_{p(\boldsymbol{y})}\text{prec}_n\right)^{w_n}.
\tag{7}
$$

The approximation, though somewhat arbitrary, is necessary for efficient and tractable computation, and we found the resulting metric still correlates well with the original BLEU, as shown in our experiments. We optimize the right-hand side approximated objective in the following, where

$$
\mathbb{E}_{p(\boldsymbol{y})}\text{prec}_n = \frac{1}{T-n+1} \sum\nolimits_{i=1}^{T-n+1} \mathbb{E}_{p(\boldsymbol{y})}\left[o_{n,i}\right].
\tag{8}
$$

Recall that (as defined in Eqs.3-4):

$$
o_{n,i} = \min\left(1, \frac{v^*_{n,i}}{v_{n,i}}\right) \quad , \quad v^*_{n,i} = \sum\nolimits_{j'=1}^{T^*-n+1} \mathbb{1}[\boldsymbol{y}^*_{j':j'+n} = \boldsymbol{y}_{i:i+n}].
\tag{9}
$$

That is, for each pair $(n,i)$, the quantity $v^*_{n,i}$, and thus $o_{n,i}$, is *non-zero* only if the sub-sequence $\boldsymbol{y}_{i:i+n}$ occurs in the reference sequence $\boldsymbol{y}^*$ (so that $\mathbb{1}[\boldsymbol{y}^*_{j':j'+n} = \boldsymbol{y}_{i:i+n}] = 1$ for some $j'$), namely, $\boldsymbol{y}_{i:i+n} \in \text{gram}_n(\boldsymbol{y}^*)$.

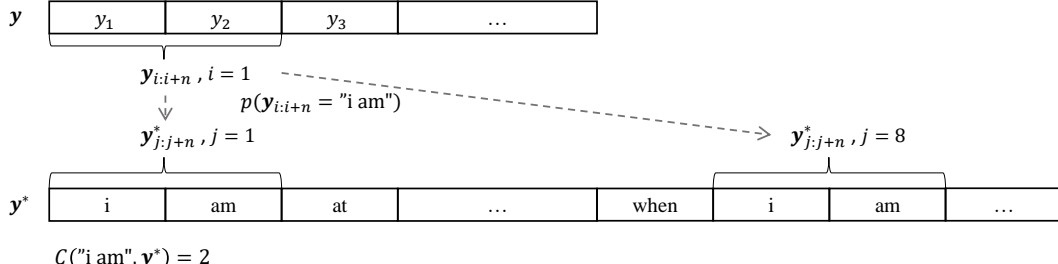

Figure 1: An example value of $\boldsymbol{y}_{i:i+n}$ in Eq.(10). As described in the text, effective values for $\boldsymbol{y}_{i,i+n}$ that can contribute to the final BLEU score are the set of $n$-grams in the reference $\boldsymbol{y}^*$. Here, $i = 1$, $n = 2$, and $\boldsymbol{y}_{i,i+n}$ takes value of "*i am*" which occurs twice in $\boldsymbol{y}^*$ ($j = 1$ and $j = 8$). The probability $p(\boldsymbol{y}_{i,i+n} = $ "i am") is thus counted twice when enumerating $j$, and hence in Eq.(10) we divide $C($"i am", $\boldsymbol{y}^*)$ to avoid such duplicate count.

With the key observation, for each term $E_{p(\boldsymbol{y})}[o_{n,i}]$ of Eq.(8), we decompose $\boldsymbol{y}$ into $\boldsymbol{y}_{i:i+n}$ and the remaining $\boldsymbol{y}_{\neg i:i+n}$[1], and explicitly enumerate all effective values of $\boldsymbol{y}_{i:i+n}$ by simply enumerating the $n$-grams of $\boldsymbol{y}^*$:

$$
\begin{aligned}
\mathbb{E}_{p(\boldsymbol{y})}\left[o_{n,i}\right] &= \mathbb{E}_{p(\boldsymbol{y}_{i:i+n})}\mathbb{E}_{p(\boldsymbol{y}_{\neg i:i+n})}\left[o_{n,i}\right] \\
&= \sum\nolimits_{\boldsymbol{s}\in\mathrm{gram}_n(\boldsymbol{y}^*)} p(\boldsymbol{y}_{i:i+n} = \boldsymbol{s}) \cdot \mathbb{E}_{p(\boldsymbol{y}_{\neg i:i+n})}\left[o_{n,i}\right] \\
&= \sum\nolimits_{j=1}^{T^*-n+1} \frac{1}{C(\boldsymbol{y}^*_{j:j+n}, \boldsymbol{y}^*)} \cdot p(\boldsymbol{y}_{i:i+n} = \boldsymbol{y}^*_{j:j+n}) \cdot \mathbb{E}_{p(\boldsymbol{y}_{\neg i:i+n})}\left[o_{n,i}\right].
\end{aligned}
\tag{10}
$$

The third equation enumerates the starting index $j$ of $n$-grams in $\boldsymbol{y}^*$, which necessitates to divide the occurrence time of $\boldsymbol{y}^*_{j:j+n}$, namely $C(\boldsymbol{y}^*_{j:j+n}, \boldsymbol{y}^*)$, to avoid duplicate count. Figure 1 illustrate an example of enumerating $j$.

The only difficult part above is the last term $\mathbb{E}_{p(\boldsymbol{y}_{\neg i:i+n})}[o_{n,i}]$. For computational tractability, we make the following approximations:

$$
\mathbb{E}_{p(\boldsymbol{y}_{\neg i:i+n})}\left[o_{n,i}\right] = \mathbb{E}_{p(\boldsymbol{y}_{\neg i:i+n})}\min\left(1, \frac{v^*_{n,i}}{v_{n,i}}\right) \approx \min\left(1, \mathbb{E}_{p(\boldsymbol{y}_{\neg i:i+n})}\frac{v^*_{n,i}}{v_{n,i}}\right) \approx \min\left(1, \frac{v^*_{n,i}}{\mathbb{E}_{p(\boldsymbol{y}_{\neg i:i+n})}v_{n,i}}\right),
\tag{11}
$$

where the first equation is by definition of $o_{n,i}$ (Eq.9); the first approximation is due to the exchange of the expectation operation with the $\min(\cdot, \cdot)$ function; and the second approximation stems from applying the expectation directly to the denominator. Note that the numerator $v^*_{n,t}$ is independent of $\boldsymbol{y}_{\neg i:i+n}$: by the definition in Eq.(3) and the condition $\boldsymbol{y}_{i:i+n} = \boldsymbol{y}^*_{j:j+n}$ from Eq.(10), we have $v^*_{n,i} = C(\boldsymbol{y}^*_{j:j+n}, \boldsymbol{y}^*)$.

The last intractability for our BLEU reformulation is to compute the denominator in Eq.(11). By definition of $v_{n,i}$ (Eq.3), we have:

$$
\begin{aligned}
\mathbb{E}_{p(\boldsymbol{y}_{\neg i:i+n})}v_{n,i} &= \sum\nolimits_{i'=1}^{T-n+1} \mathbb{E}_{p(\boldsymbol{y}_{\neg i:i+n})}\mathbb{1}\left[\boldsymbol{y}_{i':i'+n} = \boldsymbol{y}_{i:i+n}\right] \\
&= \sum\nolimits_{i'=1}^{T-n+1} \mathbb{E}_{p(\boldsymbol{y}_{\neg i:i+n})}\mathbb{1}\left[\boldsymbol{y}_{i':i'+n} = \boldsymbol{y}^*_{j:j+n}\right],
\end{aligned}
\tag{12}
$$

where the second equation is because $\boldsymbol{y}_{i:i+n}$ has taken value of $\boldsymbol{y}^*_{j:j+n}$ in Eq.(10). We consider all three cases for the pair $\boldsymbol{y}_{i':i'+n}$ and $\boldsymbol{y}_{i:i+n}$:

1) $\boldsymbol{y}_{i':i'+n}$ refers to the same $n$-gram sub-sequence as $\boldsymbol{y}_{i:i+n}$ (i.e., $i' = i$). It is clear that $\mathbb{E}_{p(\boldsymbol{y}_{\neg i:i+n})}\mathbb{1}\left[\boldsymbol{y}_{i':i'+n} = \boldsymbol{y}^*_{j:j+n}\right] = 1$.

2) $\boldsymbol{y}_{i':i'+n}$ does not overlap with $\boldsymbol{y}_{i:i+n}$ (i.e., $|i' - i| \geq n$), which means $\boldsymbol{y}_{i':i'+n}$ is independent of $\boldsymbol{y}_{i:i+n}$, and thus $\mathbb{E}_{p(\boldsymbol{y}_{\neg i:i+n})}\mathbb{1}\left[\boldsymbol{y}_{i':i'+n} = \boldsymbol{y}^*_{j:j+n}\right] = p(\boldsymbol{y}_{i':i'+n} = \boldsymbol{y}^*_{j:j+n})$.

---

[1]Here we assume $\boldsymbol{y}_{i:i+n}$ and $\boldsymbol{y}_{\neg i:i+n}$ are independent, so that $p(\boldsymbol{y})$ is decomposed as $p(\boldsymbol{y}) = p(\boldsymbol{y}_{i:i+n})\,p(\boldsymbol{y}_{\neg i:i+n})$.

3) Part of $\boldsymbol{y}_{i':i'+n}$ overlaps with $\boldsymbol{y}_{i:i+n}$ (i.e., $0 < |i' - i| < n$). In this case, only the non-overlapping part of $\boldsymbol{y}_{i':i'+n}$ is random variable to be marginalized out. Thus, differing from case 2), we generally have $\mathbb{E}_{p(\boldsymbol{y}_{\neg i:i+n})}\mathbb{1}[\boldsymbol{y}_{i':i'+n} = \boldsymbol{y}^*_{j:j+n}] \geq p(\boldsymbol{y}_{i':i'+n} = \boldsymbol{y}^*_{j:j+n})$. However, for computational simplicity, we simply use the latter for approximation.

With the above discussion, Eq.(12) is approximated as:

$$\mathbb{E}_{p(\boldsymbol{y}_{\neg i:i+n})}v_{n,i} \approx 1 + \sum_{\substack{i'=1 \\ i' \neq i}}^{T-n+1} p(\boldsymbol{y}_{i':i'+n} = \boldsymbol{y}^*_{j:j+n}) \tag{13}$$

**Summary**

We have completed the BLEU reformulation. In particular, Eq.(10) made the key step that identifies the small set of effective values for each $\boldsymbol{y}_{i:i+n}$, which is exactly the set of $n$-grams of the reference $\boldsymbol{y}^*$. As $\boldsymbol{y}^*$ is given, direct enumeration of its $n$-grams is straightforward and computationally efficient.

More specifically, combining Eqs.(10,11,13), we can approximate as:

$$\mathbb{E}_{p_\theta(\boldsymbol{y})}[o_{n,i}] \approx \sum_{j=1}^{T^*-n+1} \frac{p_\theta(\boldsymbol{y}_{i:i+n} = \boldsymbol{y}^*_{j:j+n})}{C(\boldsymbol{y}^*_{j:j+n}, \boldsymbol{y}^*)} \min\left(1, \frac{C(\boldsymbol{y}^*_{j:j+n}, \boldsymbol{y}^*)}{1 + \sum_{\substack{i'=1 \\ i' \neq i}}^{T-n+1} p_\theta(\boldsymbol{y}_{i':i'+n} = \boldsymbol{y}^*_{j:j+n})}\right), \tag{14}$$

$$\triangleq \tilde{o}_{n,i}$$

where the model distribution $p_\theta$ is invoked only for evaluating the likelihood of given reference $n$-grams $\boldsymbol{y}^*_{j:j+n}$[2]. We discuss the implementation of the likelihood evaluation in the next section. Note that there is no need of stochastic sampling from the huge sequence space as in the original policy gradient expected BLEU objective (Eq.5). The *gradient* of Eq.(14) w.r.t. $\boldsymbol{\theta}$ can also be straightforwardly computed. (The $\min(\cdot, \cdot)$ operation may invoke subgradient, which is minor in practice.)

Plugging Eq.(14) into Eq.(8), we obtain

$$\mathbb{E}_{p(\boldsymbol{y})}\text{prec}_n \approx \frac{1}{T-n+1}\sum_{i=1}^{T-n+1} \tilde{o}_{n,i} \triangleq \widetilde{\text{prec}}_n, \tag{15}$$

and further plugging the above into Eq.(7), we obtain the full, approximated reformulation of the expected BLEU objective:

$$\mathbb{E}_{p(\boldsymbol{y})}\text{BLEU} \approx \text{BP} \cdot \prod_{n=1}^{N} \widetilde{\text{prec}}_n^{w_n}. \tag{16}$$

In practice, we found it is more stable and simple-to-implement by maximizing the *logarithm* of the resulting formulation. We thus define the final DEBLEU objective as:

$$\mathcal{L}_{\text{DEBLEU}} \triangleq -\log\text{BP} - \sum_{n=1}^{N} w_n \log \widetilde{\text{prec}}_n \tag{17}$$

In summary, Eqs.(17,15,14) fully define the proposed DEBLEU objective, which is fully differentiable w.r.t. the model parameter $\boldsymbol{\theta}$ and is therefore end-to-end trainable.

### 3.3 TRAINING & IMPLEMENTATION

We now discuss the implementation and training process of the proposed DEBLEU objective. In particular, we devise a simple mask-and-anneal procedure that optimizes the objective smoothly. We further analyze the computational complexity of the objective, showing that the computation is efficient, comparable to the common cross-entropy objective.

**Gumbel-softmax Decoding with Teacher Masks**

Recall that for sequence generation models such as recurrent networks, we have the step-wise decomposition of the sequence distribution $p_\theta(\boldsymbol{y}) = \prod_i p_\theta(y_i \mid \boldsymbol{y}_{1:i})$. The main computation of the DEBLEU objective is to evaluate the likelihood $p_\theta(\boldsymbol{y}_{i:i+n} = \boldsymbol{y}^*_{j:j+n})$ for each $i$, $j$, and $n$. To this

---

[2]Interestingly, recall that in the common maximum-likelihood learning with cross-entropy loss, the model distribution $p_\theta$ is invoked for evaluating the likelihood of the whole reference sequence, i.e., $p(\boldsymbol{y} = \boldsymbol{y}^*)$.

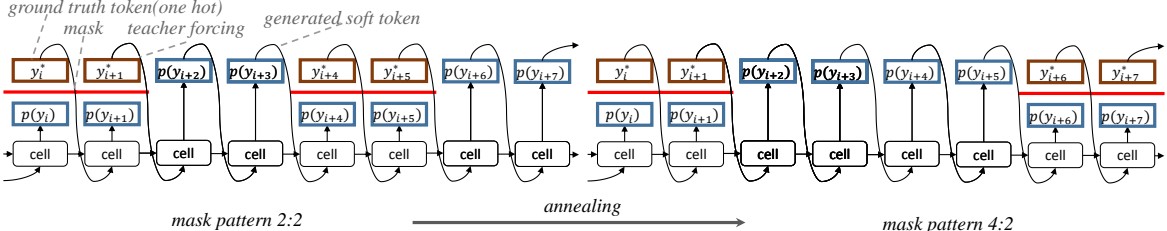

Figure 2: An illustration of decoding with teacher masks. The red lines denote masked steps, for which the corresponding one-hot ground-truth token is used for both DEBLEU evaluation and next step decoding. For unmasked steps, the output Gumbel-softmax distribution is used as a soft token and fed to the next step. Left panel illustrates a mask pattern of *2:2*, and the right panel illustrates a *4:2* pattern. As the training proceeds, annealing between the patterns is performed. See section 3.3 for more details.

end, we first perform *Gumbel-softmax* decoding to obtain the step-wise distribution $p_\theta(y_i \mid \boldsymbol{y}_{1:i})$ for each step, and compute the likelihood by multiplying the probabilities of relevant steps, namely $p_\theta(\boldsymbol{y}_{i:i+n} = \boldsymbol{y}^*_{j:j+n}) = \prod_{t=0}^{n-1} p_\theta(y_{i+t} = y^*_{j+t} \mid \boldsymbol{y}_{1:i+t})$. As a popular decoding approach (Jang et al., 2016; Hu et al., 2017), Gumbel-softmax decoding at each step feeds the output distribution to the next step as a *soft* input token (See Figure 2 for an example).

In practice, however, we found using the above decoded $p_\theta(\boldsymbol{y}_{i:i+n} = \boldsymbol{y}^*_{j:j+n})$ for every step $i$ can lead to unstable results especially at the early stage of training. This is partially because of the accumulated error of using probabilities to replace the hard count in the original BLEU (see, e.g., Eq.12). To address the issue, we introduce *teacher masks*. That is, for a set of selected steps $i$, we replace the distribution $p_\theta(y_i \mid \boldsymbol{y}_{1:i})$ with the respective one-hot representation of ground-truth token $y^*_i$ (i.e., the distribution used is now $p(y_i) = 1$ if $y_i = y^*_i$ and 0 otherwise). Figure 2 illustrates some examples of such masked steps. We found such replacement can make the evaluation of the DEBLEU and its gradient more stable. Besides, for a masked step, we also use the one-hot ground-truth token as the input to the next step (Figure 2). This resembles the teacher-forcing decoding used in vanilla maximum-likelihood learning, and hence the name "teacher mask".

We interleave masked and unmasked steps. Specifically, given a mask pattern *#unmasked:#masked*, we apply a mask such that *#unmasked* consecutive steps are not masked, followed with *#masked* consecutive steps that are masked. For example, typical mask patterns are *2:2* (Figure 2, left panel), *4:2* (Figure 2, right panel), and *1:0* (i.e., no mask). Such regular-shaped (as opposed to randomly-sampled) masks correspond to the characteristics of (DE)BLEU evaluation in which consecutive steps are usually grouped together to form an $n$-gram and compute respective likelihood. Note that the masks also implement certain randomness across training iterations by shifting to left or right for a random number of steps.

As the training proceeds, we *anneal* the mask pattern by gradually increasing the portion of unmasked steps. For example, after the model converges with a mask pattern of *2:2*, we change to apply a pattern of *4:2*. The annealing continues until no masks are used. This in effect creates a curriculum learning (Bengio et al., 2009) strategy that gradually increases the difficulty of the optimization problem.

**Pretraining**   In practice we first pretrain the model by minimizing the vanilla maximum-likelihood cross-entropy loss, and continue to train the model with the proposed DEBLEU objective using the above mask-and-anneal procedure.

### Complexity Analysis

As above, the main computation of DEBLEU falls in evaluating the likelihood $p_\theta(\boldsymbol{y}_{i:i+n} = \boldsymbol{y}^*_{j:j+n})$ for $i \in \{1, \dots, T\}$, $j \in \{1, \dots, T^*\}$, and $n \in \{1, \dots, N\}$. The values can be efficiently computed with a complexity of $O(N \cdot T \cdot T^* + V \cdot T)$ (see Appendix A for details). In comparison, the common maximum likelihood cross-entropy loss has a computational complexity of $O(V \cdot T)$. Since in practice $V \cdot T$ usually dominates $N \cdot T \cdot T^*$, the proposed DEBLEU objective adds only negligible

| Method | BLEU on de-en | BLEU on en-fr |
|---|---|---|
| Cross Entropy | 22.98 | 38.37 |
| Policy Gradient | 23.24 | 38.81 |
| DEBLEU | **24.37** | **39.79** |

Table 1: BLEU scores on the German-to-English (de-en) and English-to-French (en-fr) test sets.

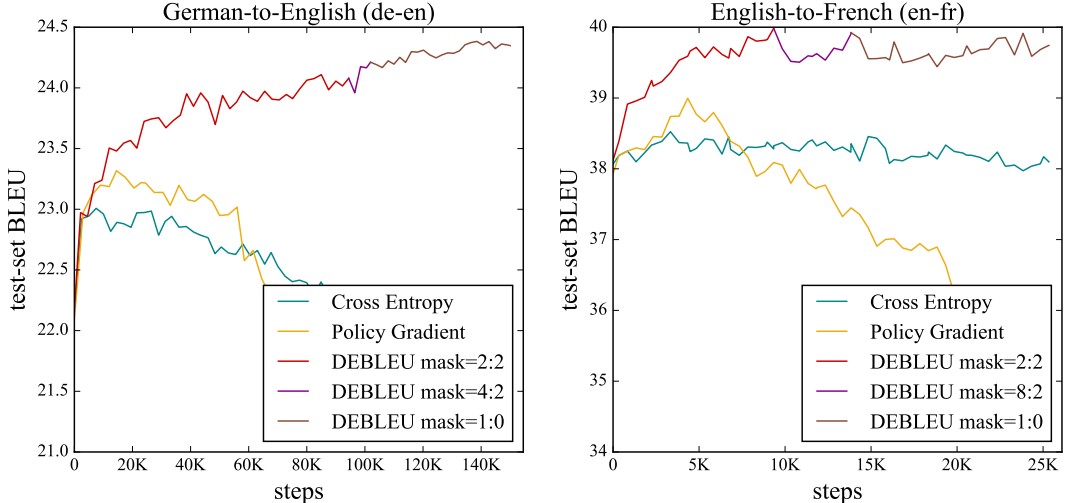

Figure 3: The curves of test-set BLEU score when training on the German-to-English (de-en) and English-to-French (en-fr) datasets, respectively. The starting point (step=0) is the model after pretraining.

computational overhead compared to the cross-entropy objective, while providing greatly improved empirical performance as shown in the next section. At last, it is worth noting that all the computation can be executed in a batch mode just as other objectives such as cross entropy.

## 4 EXPERIMENTS

We evaluate the proposed DEBLEU objective in the tasks of machine translation and image captioning. The empirical results show our method provides better performance in terms of test-set BLEU, in comparison to the popular algorithms including cross-entropy based maximum-likelihood learning as well as the policy gradient algorithm.

Throughout the experiments, we set the $n$-gram precision weights in DEBLEU (Eq.17) to $w_1 = 0.1$ and $w_2 = w_3 = w_4 = 0.3$. We found such a smaller weight of uni-gram precision helps with more stable convergence. We conjecture this is because the uni-gram precision completely ignores word order, which can, to some extent, cause instability in training.

We will release all experimental code for reproducibility upon acceptance.

### 4.1 NEURAL MACHINE TRANSLATION

**Setup**  We use both the German-to-English (de-en) and English-to-French (en-fr) datasets from IWSLT 2014 (Cettolo et al., 2014). Each dataset contains around 172K training instances. We pruned the vocabulary size of each language to around 15K. We use a sequence-to-sequence model with attention (Bahdanau et al., 2014). The encoder is a two-layer bi-directional LSTM RNN, while the decoder is a single-layer uni-directional LSTM RNN. Both the encoder and decoder have a hidden size of 1000 and a word embedding size of 500. We use the Adam SGD optimizer with learning rate annealing from $10^{-3}$ to $10^{-5}$. Both the policy gradient and the DEBLEU algorithms start with a cross-entropy pretrained model. After pretraining, the policy gradient objective is mixed with the

| Method | BLEU | ROUGE-L | METEOR | CIDEr | SPICE |
|---|---|---|---|---|---|
| Cross Entropy | 27.89 | 51.93 | 23.97 | 88.77 | 16.99 |
| Policy Gradient | 31.15 | 52.79 | 24.13 | **92.61** | 16.91 |
| DEBLEU | **31.39** | **53.37** | **24.52** | 91.49 | **17.33** |

Table 2: Image captioning results. Policy gradient optimizes the expected BLEU. The DEBLEU training achieves the best performance on most of the metrics.

cross-entropy objective with weights $0.3 : 1.0$ to obtain the best results (otherwise the performance drops quickly; we also tried other combination methods such as (Ranzato et al., 2015) but did not get better results). We further use sample decoding results averaged over 10 runs as the baseline for policy gradient. For the DEBLEU training, we anneal the teacher mask pattern from *2:2*, *4:2* to *1:0* on the German-to-English dataset, while annealing from *2:2*, *8:2* to *1:0* on the English-to-French dataset. Following previous work (Ranzato et al., 2015), at test time we use greedy decoding for evaluating the test-set BLEU. More experimental settings are provided in Appendix B.

**Results** The results of test-set BLEU scores are presented in Table 1. We can see that the proposed DEBLEU training provides significantly better performance than both the cross entropy and policy gradient training. DEBLEU improves over cross entropy as DEBLEU better correlates to the BLEU metric, closing the training/test discrepancy in the cross entropy method. The DEBLEU objective involves approximations to the expected BLEU objective optimized by policy gradient, but still yields superior results. This is because DEBLEU avoids sampling from the huge sequence space and is much easier to optimize.

We further visualize the test-set BLEU curves during training. We can see that with DEBLEU training the BLEU score increases smoothly and keeps stable after convergence. In contrast, the BLEU score drops in both the cross entropy and policy gradient cases, partially because of the misalignment between cross entropy and BLEU, and the instability of policy gradient updates. We provide some generated samples on the German-to-English (de-en) test set in the supplementary materials.

## 4.2 IMAGE CAPTIONING

**Setup** For image captioning, we use the MSCOCO dataset (Lin et al., 2014) and take the train/dev/test split from (Karpathy & Fei-Fei, 2015). We follow (Vinyals et al., 2015) for data preprocessing and model setup. In particular, a pretrained 101-layer ResNet (He et al., 2016) is used to encode images into feature vectors as inputs to the decoder. The encoder is fixed throughout training. The decoder is a single-layer LSTM RNN with both the hidden size and embedding size set to 512. All other settings are similar to those in machine translation. Please see Appendix B for more details.

**Results** Table 2 shows the image captioning results, including the test-set BLEU score of interest as well as other popular evaluation metrics. As in machine translation, DEBLEU performs best in terms of the test-set BLEU metric. It is interesting to see that DEBLEU also achieves improved performance on most of other metrics, though they are not directly optimized. This validates that the proposed DEBLEU training does not merely overfit to the BLEU score, but instead improves the text generation results in general.

## 5 CONCLUSION

We have developed a new differentiable expected BLEU (DEBLEU) objective for end-to-end training of neural text generation models with direct gradient descent. The proposed method addresses the train/test discrepancy issue of common cross-entropy training, while is more efficient and practical than the policy gradient algorithm. Experiments on neural machine translation and image captioning demonstrate the superiority of the objective and training method. In our derivation, we have leveraged the decomposability and sparsity of BLEU. We believe such intuition also applies to other widely-used metrics such as ROUGE and others. We would like to explore differentiable variants of these metrics in the future.

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

## APPENDIX A    IMPLEMENTATION

The input of our DEBLEU module is (masked) $p(\boldsymbol{y})$ and the reference sequence $\boldsymbol{y}^* = (y_1^*, \ldots, y_{T^*}^*)$. $p(\boldsymbol{y})$ can be represented as a $T \times V$ matrix, denoted by $P$. Then we apply Eqs.(14,15,17) sequentially. The major computation is in Eq.(14), where all values of $p(\boldsymbol{y}_{i:i+n} = \boldsymbol{y}_{j:j+n}^*)$ and $C(\boldsymbol{y}_{j:j+n}^*, \boldsymbol{y}^*)$ are required.

First we will show how to obtain $p(\boldsymbol{y}_{i:i+n} = \boldsymbol{y}_{j:j+n}^*)$. Based on our independence assumption again, we decompose the probability:

$$p(\boldsymbol{y}_{i:i+n} = \boldsymbol{y}_{j:j+n}^*) = \prod_{d=0}^{n-1} p(y_{i+d} = y_{j+d}^*) \tag{18}$$

$p(y_i = y_j^*), \forall i, j$ is now required. Simply $p(y_i = y_j^*) = P_{i, y_j^*}$. Thus, all values can be obtained by an index selection operation of $P$ with $\boldsymbol{y}^*$ as indexes. We denote the result as matrix $M \in \mathbb{R}^{\tilde{T} \times T^*}$, in which $M_{i,j} = p(y_i = y_j^*)$. Then, we would like to obtain $M^n \in \mathbb{R}^{(\tilde{T}-n+1) \times (T^*-n+1)}$, in which $M_{i,j}^n = p(\boldsymbol{y}_{i:i+n} = \boldsymbol{y}_{j:j+n}^*), n \in \{1, \ldots, N\}$. Obviously, $M^1 = M$. For $n > 1$, we utilize $M^{n-1}$ to save computation. Notice that

$$p(\boldsymbol{y}_{i:i+n} = \boldsymbol{y}_{j:j+n}^*) = p(y_i = y_j^*)\, p(\boldsymbol{y}_{i+1:i+n} = \boldsymbol{y}_{j+1:j+n}^*) \tag{19}$$

which implies that

$$M^n = M \circ M_{2:,2:}^{n-1} \tag{20}$$

where $M_{2:,2:}^{n-1}$ denotes $M^{n-1}$ removed the first row and column and $\circ$ denotes the Hadamard (element-wise) product.

As for $C(\boldsymbol{y}_{j:j+n}^*, \boldsymbol{y}^*)$, we can also obtain it in the same way. We can even preprocess and store it before training. Anyway, it is not bottleneck our computation.

**Batch training**    In order to utilize the parallelism of the computing devices, batch training is usually used. All computation above can be batchized. The only thing to care about is the padding. The target sequences are usually padded to the same length, but padding shall not be regarded as real tokens in computation. Therefore, we mask out the padding part in matrices $P$ and $M^1, \ldots, M^N$.

## APPENDIX B    EXPERIMENTAL SETTINGS

**Machine Translation**    For dataset preprocessing: In the de-en dataset, we remove all punctuations. In en-fr dataset, we lowercase all characters, make the punctuation as tokens, and separate words at apostrophes or hyphens.

For test sets: The test set of the de-en task is created by merging all test sets on IWSLT 2014 official site; The test set of en-fr is the tst2012 set (see Cettolo et al. (2014) for more details).

**Image Captioning**    The decoder the model is a 1-layer LSTM. The hidden size of the LSTM decoder and the embedding size are set to 512. And 50% dropout is applied on the decoder. Following common online implementations, during pretraining, learning rate decay and scheduled sampling are applied. After pretraining, all learning rate decay and scheduled sampling are removed. The learning rate is fixed afterwards.

## APPENDIX C    GENERATED SENTENCE EXAMPLES

|  | **Generated Sentence** |
|---|---|
| Reference | Well today we know everything about where our objects come from |
| Cross Entropy | Now today we know everything that come from |
| DEBLEU | Now today we know everything about where our things come from |
| Reference | Let's get half of us to agree to spend an hour a day playing games until we solve real world problems |
| Cross Entropy | We should be able to <UNK> that half of us spend an hour per day with games until we have the problems of the real world |
| DEBLEU | We should <UNK> that half of us spend an hour a day spend an hour until we solved the problems of the real world |
| Reference | So what you suddenly started to realize or what I started to realize is that when you started having conversations with these companies the idea of understanding your brand is a universal problem |
| Cross Entropy | So what you suddenly began to understand or what I started to understand was that when you start talking to these companies the idea of how your brand is understood is a <UNK> problem |
| DEBLEU | So what you suddenly started to understand or what I started to understand was that if you start talking to these companies the idea of how your brand is understood a <UNK> problem |
| Reference | And I always tell people that I don't want to show up looking like a scientist |
| Cross Entropy | And I always tell people that I don't want to get a scientist like this |
| DEBLEU | And I say people always I don't want to get like a scientist |

Table 3: Examples of generated sentences on the IWSLT 2014 German-to-English (de-en) test set.

