# OpenReview forum: "Differentiable Expected BLEU for Text Generation"
_ICLR.cc/2019/Conference_

### Official Review · AnonReviewer2 · 2018-11-01

**Rating:** 6
**Confidence:** 4

**Review:**

The paper describes a differentiable expected BLEU objective which computes expected n-gram precision values by ignoring the brevity penalty.

Clarity:
Section 3 of the paper is very technical and hard to follow. Please rewrite this section to be more accessible to a wider audience by including diagrams and more explanation.

Originality/signifiance: the idea of making BLEU differentiable is a much researched topic and this paper provides a nice idea on how to make this work.

Evaluation:
The evaluation is not very strong for the following reasons:

1) The IWSLT baselines are very weak. For example, current ICLR submissions, report cross-entropy baselines of >33 BLEU, whereas this paper starts from 23 BLEU on IWSTL14 de-en (e.g., https://openreview.net/pdf?id=r1gGpjActQ), even two years ago baselines were stronger: https://arxiv.org/abs/1606.02960

2) Why is policy gradient not better? You report a 0.26 BLEU improvement on IWSLT de-en, which is tiny compared to what other papers achieved, e.g., https://arxiv.org/abs/1606.02960, https://arxiv.org/abs/1711.04956

3) The experiments are on some of the smallest translation tasks. IWSLT is very small and given that the method is supposed to be lightweight, i.e., not much more costly than cross-entropy, it should be feasibile to run experiments on larger datasets.

This makes me wonder how significant any improvements would be with a good baseline and on a larger datasets.

Also, which test set are you using?

Finally, in Figure 3, why is cross-entropy getting worse after only ~2-4K updates? Are you overfitting?
Please reference this figure in the text.

---

### Official Review · AnonReviewer1 · 2018-11-02
**The paper misses important references. It chooses an empirical setup which prevents comparison with related work, and the report results on de-en seem weak. The proposed approach does not bound or estimate how far from BLEU is the proposed approximation. This means that the authors need to justify empirically that it preserves correlation with BLEU, which is not shown in the paper.**

**Rating:** 4
**Confidence:** 5

**Review:**

Differentiable Expected BLEU for Text Generation

Paper Summary:

Neural translation systems optimizes training data likelihood, not the end metric of interest BLEU. This work proposes to approximate BLEU with a continuous, differentiable function that can be optimized during training.

Review:

The paper reads well. It has a few but crucial missing references. The motivation is easy to understand and a relevant problem to work on. The main weaknesses of the work lies in its very loose derivations, and its weak empirical results.

First on context/missing references: the author ignores approaches optimizing BLEU with log linear models (Franz Och 2003), and the structured prediction literature in general, both for exact (Tsochantaridis et al 2004) and approximate search (Daume and Marcu 2005). This type of approach has been applied to NMT recently (Edunov et al 2018). Your paper also misses important references addressing BLEU optimization with reinforcement strategies (Norouzi et al 2016) or (Bahdanau et al 2017). Although not targeting BLEU directly (Wiseman and Rush 16) is also a reference to cite wrt optimizing search quality directly.

On empirical results, you chose to work IWSLT in the de-en direction while most of the literature worked on en-de. It prevents comparing your results to other papers. I would suggest to switch directions and/or to report results from other methods (Ranzato et al 2015; Wiseman and Rush 2016; Norouzi et al 2016; Edunov et al 2018). De-en is generally easier than en-de (generating German) and your BLEU scores are particularly low < 25 for de-en while other methods ranges in 26-33 BLEU for en-de (Edunov et al 2018).

On the method itself, approximating BLEU with a continuous function is not easy and the approach you take involves swapping function composition and expectation multiple times in a loose way. You acknowledge that (7) is unprincipled but (10) is also problematic since this equation does not acknowledge that successive ngrams overlap and cannot be considered independent. Also, the dependence of successive words is core to NMT/conditional language models and the independence hypothesis from the footnote on page 4 can be true only for a bag of word model. Overall, I feel that given the shortcuts you take, you need to justify that your approximation of BLEU is still correlated with BLEU. I would suggest to sample from a well trained NMT system to collect several hypotheses and to measure how well your BLEU approximation correlate with BLEU. How many times BLEU decides that hypA > hypB but your approximation invert this relation? is it true for large difference, small difference of BLEU score? at low BLEU score, high BLEU score?

Finally, you do not mention the distinction between expected BLEU  \sum_y P(y|x) BLEU(y, ref) and the BLEU obtained by beam search which only look at (an estimate of) the most likely sequence y* = argmax P(y|x) . Your approach and most reinforcement strategy targets optimizing expected BLEU, but this has no guarantee to make BLEU(y*, ref) any better. Could you report both an estimate of expected BLEU and beam BLEU for different methods? In particular, MERT (), beam optimization (Wiseman and Rush 2016) and  structured prediction (Edunov et al 2018) explicitly make this distinction. This is not a side issue as this discussion is in tension with your motivations.

Review Summary:

The paper misses important references. It chooses an empirical setup which prevents comparison with related work, and the report results on de-en seem weak. The proposed approach does not bound or estimate how far from BLEU is the proposed approximation. This means that the authors need to justify empirically that it preserves correlation with BLEU, which is not shown in the paper.

Missing references

An Actor-Critic Algorithm for Sequence Prediction (ICLR 2017)  Dzmitry Bahdanau, Philemon Brakel, Kelvin Xu, Anirudh Goyal, Ryan Lowe, Joelle Pineau, Aaron Courville, Yoshua Bengio

Hal Daume III and Daniel Marcu. Learning as search optimization: Approximate large margin methods for structured prediction. ICML 2005.

Sergey Edunov, Myle Ott, Michael Auli, David Grangier, Marc'Aurelio Ranzato
Classical Structured Prediction Losses for Sequence to Sequence Learning, NAACL 18

Minimum Error Rate Training in Statistical Machine Translation Franz Josef Och. 2003 ACL

I. Tsochantaridis, T. Hofmann, T. Joachims, and Y. Altun, Support Vector Machine Learning for Interdependent and Structured Output Spaces, ICML 2004.

Mohammad Norouzi, Samy Bengio, Zhifeng Chen, Navdeep Jaitly, Mike Schuster, Yonghui Wu, Dale Schuurmans, Reward Augmented Maximum Likelihood for Neural Structured Prediction, 2016

Sequence-to-Sequence Learning as Beam-Search Optimization, Sam Wiseman and Alexander M. Rush., EMNLP 2016

---

### Official Review · AnonReviewer3 · 2018-11-03
**Too many approximations in formulation, few experiments and discussion**

**Rating:** 4
**Confidence:** 4

**Review:**

This paper proposed a differentiable metric for text generation tasks inspired by BLEU and a random training method by the Gumbel-softmax trick to utilize the proposed metric. Experiments showed that the proposed method improves BLEU compared with simple cross entropy training and policy gradient training.

Pros:
* The new metric provides a direct perspective on how good the conjunction of a generated sentence is, which has not been provided other metric historically used on language generation tasks, such as cross-entropy.

Cons:
* Too many approximations that blur the relationship between the original metric (BLEU) and the derived metric.
* Not enough experiments and poor discussion. Authors should consume more space in the paper for experiments.

The formulation of the metric consists of many approximations and it looks no longer BLEU, although the new metric shares the same motivation: "introducing accuracy of n-gram conjunction" to evaluate outputs. Selecting BLEU as the starting point of this study seems not a reasonable idea. Most approximations look gratuitously introduced to force to modify BLEU to the final metric, but choosing an appropriate motivation first may conduct more straightforward metric for this purpose.

In experiments on machine translation, its setting looks problematic. The corpus size is relatively smaller than other standard tasks (e.g., WMT) but the size of the network layers is large. This may result in an over-fitting of the model easily, as shown in the results of cross-entropy training in Figure 3. Authors mentioned that this tendency is caused by the "misalignment between cross entropy and BLEU," however they should first remove other trivial reasons before referring an additional hypothesis.
In addition, the paper proposed a training method based on Gumbel softmax and annealing which affect the training stability through additional hyperparameters and annealing settings. Since the paper provided only one training case of the proposed method, we couldn't discuss if the result can be generalized or just a lucky.

If the lengths of source and target are assumed as same, the BP factor becomes always 1. Why the final metric (Eq. 17) maintains this factor?

---

### Public Comment · (anonymous) · 2018-10-23
**Difference between DEBLEU and LB of [1]? Novelty and results of experiments on IWSLT'14?**

Hi!

Comparing with [1], derivation of DEBLEU objective looks very similar to "lower bound" (LB) in [1] (it should be noted however, that your derivation is much easier to follow). What is a general difference between DEBLEU and LB of [1]?

You refer [1] as "made preliminary attempts to develop differentiable approximations of BLEU for neural model training, but only studied on toy tasks", however, the latest version of [1] in arXiv (dated August 23) includes experiments on IWSLT'14 and WMT'14 datasets, which show improvement over both cross-entropy and direct BLEU objectives. Moreover, [1] reports significantly higher BLEU scores with a smaller network for all objectives.

[1] Vlad Zhukov, Eugene Golikov, Maksim Kretov - Differentiable lower bound for expected BLEU score. arXiv:1712.04708v4

---

> ### Author Response · Authors · 2018-10-27
> **Thanks and response**
>
> Thanks for your valuable comment. We didn’t notice the latest update of [1] which was released on arXiv only around one month prior to this submission. With the new results in [1], we will update our statement of “toy tasks” accordingly. We appreciate for pointing this out! As in the paper, we’ve said “our formulation uses a couple of similar approximations or assumptions” with [1] and (Casas et al., 2018). Here we emphasize the difference of our work with [1] as below:
>     * Our formulation stems from a different and clear intuition of leveraging the sparsity of BLEU score, and decomposes the goal into multiple derivation steps with clear motivations.
>     * We’ve developed a mask-and-anneal training process to stabilize the training. We also describe key implementation and analyze the computational complexity which is comparable to common cross-entropy training.
>     * Our formulation naturally leads to Gumbel-softmax decoding for differentiable BLEU training and gradient backpropagation along time steps, while in [1] it’s unclear to us what decoding strategy is used.
>     * We believe the claim of “lower bound” in [1] could be problematic. For example, in Eq.(20) in [1], the inequality does not necessarily hold since `min(1, c/x)` is not a convex function of x.
>
> The difference of experimental results on IWSLT’14 can be attributed to different data preprocessing procedures and model configurations (e.g., input to each decoding step, #layers in encoder, etc). We will release the code.

---

### Meta-Review · Area_Chair1 · 2018-12-14
**The paper needs improvement**

**Confidence:** 4
**Recommendation:** Reject

**Metareview:**

The paper presents a differentiable approximation of BLEU score, which can be directly optimized using SGD. The reviewers raised concerns about (1) direct evaluation of the quality of the approximation and (2) the significance of the experimental results. There is also a concern (3) regarding the significance of BLEU score in the first place, and whether BLEU is the right metric that one needs to directly optimize. The authors did not provide a response, and based on the concerns above (especially 1-2) I believe that the paper does not pass the bar for acceptance at ICLR.